# Factors associated with chronic kidney disease knowledge and preventive practices: An analytical cross-sectional study among patients with hypertension at Amana Regional Referral Hospital in Dar es Salaam, Tanzania

Joel Seme Ambikile[1]*, Shabani S. Ngulupi[2], Agnes Fredrick Massae[3]

1 Clinical Nursing Department, Muhimbili University of Health and Allied Sciences (MUHAS), Dar es Salaam, Tanzania, 2 Department of Internal Medicine, Amana Regional Referral Hospital, Dar es Salaam, Tanzania, 3 Communty Health Nursing Department, Muhimbili University of Health and Allied Sciences (MUHAS), Dar es Salaam, Tanzania

* joelambikile@yahoo.com

## Abstract

### Background

Globally, Chronic kidney Disease (CKD) has become a significant public health concern, with sub-Saharan Africa being among the populations experiencing the highest rates. Managing CKD poses a significant challenge due to its health complications and associated high cost of care. Hypertension is one of the leading causes of CKD, responsible for the decline in kidney function in patients. Knowledge and lifestyle modifications are pivotal to the prevention and progression of CKD. In Tanzania, little is known regarding CKD knowledge and preventive practices among patients with hypertension. Therefore, this study aimed to assess context-specific factors associated with CKD knowledge and preventive practices among patients with hypertension at Amana Regional Referral Hospital in Dar es Salaam.

### Methods

An analytical cross-sectional study involving 184 patients was conducted at Amana Regional Referral Hospital. Patient were recruited at the outpatient hypertension clinic using simple random sampling in June and July 2022. Data analysis was conducted using IBM-SPSS Statistics version 25. Bivariate and multiple logistic regression analyses were performed to identify factors associated with knowledge and preventive practices related to CKD, with statistical significance set at a p-value < 0.05.

### Results

The median CKD knowledge score was 13 (IQR: 9–16). Of the 184 respondents, 104 (56.5%) demonstrated high CKD knowledge. None of the sociodemographic factors were significantly associated with CKD knowledge. The median score for CKD preventive practices was 9 (IQR: 8–9), and 111 respondents (60.3%) had good practices. High CKD

**Data Availability Statement:** All relevant data are within the manuscript and its Supporting information files.

**Funding:** This study was funded by the Ministry of Health in Tanzania. The funders had no role in study design, data collection and analysis, decision to publish, or preparation of the manuscript.

**Competing interests:** The authors have declared that no competing interests exist.

knowledge was significantly associated with good CKD preventive practices (AOR: 1.98; 95% CI: 1.08, 3.62; $p$ = 0.027).

## Conclusion

A significant proportion of hypertensive patients in this study exhibited both high CKD knowledge and good preventive practices. The positive correlation between CKD knowledge and improved preventive practices highlights the importance of educational interventions to further enhance CKD knowledge among patients with hypertension.

## 1. Introduction

Chronic kidney disease (CKD) has emerged as a significant public health concern in recent years. Globally, it affects approximately 752.1 million individuals and is responsible for 2.1 million deaths annually [1, 2]. The prevalence of CKD varies across regions, with Asian and sub-Saharan African populations experiencing the highest rates. In Europe, the prevalence of CKD ranges from 3.3% to 10.4%, while in the Americas, it is about 10.4%. Asia sees a prevalence of approximately 33.3%, with a majority of cases found in China and India [1]. In Africa, 15.8% of the population is affected by CKD, with lower prevalence in North Africa (6.1%) and higher rates in sub-Saharan regions: South Africa (10.4%), Eastern Africa (14.4%), Middle Africa (16.0%), and Western Africa (19.8%) [3]. In Tanzania, studies have estimated the prevalence of CKD to range from 7.0% to 13.6%, varying between urban and rural areas [4].

CKD progresses through five irreversible stages, classified as stages 1, 2, 3, 4, and 5 [5]. This progression is marked by a reduction in glomerular filtration rate (GFR). When the GFR falls below 15 ml/min/1.73m$^2$, the patient is diagnosed with end-stage kidney disease (ESKD). ESKD is associated with life-threatening complications such as anemia, uremia, hyperphosphatemia, hyperkalemia, and metabolic acidosis. These complications increase the risk of cardiovascular diseases, impair patient quality of life, and contribute to higher morbidity and mortality rates [6]. Another significant challenge in managing ESKD is the high cost of care, as patients require expensive treatment modalities like dialysis or kidney transplantation [7].

Diabetes and hypertension are the leading causes of CKD in both developed and low- and middle-income countries (LMICs). Approximately 40% of all CKD cases are attributed to diabetes, while around 23% are due to hypertension [8]. These conditions not only contribute to the onset of CKD but also promote its progression from early asymptomatic stages to end-stage renal disease (ESRD), which is associated with increased healthcare costs and a reduced quality of life. The Kidney Disease Improving Global Outcome (KDIGO) 2012 guidelines recommend that patients with hypertension and diabetes be screened annually for CKD to enable early detection and slow its progression [9].

However, hypertension is a major factor responsible for the decline in kidney function in patients with both diabetic and non-diabetic kidney disease. Additionally, high blood pressure can develop early in the course of CKD and contribute to adverse outcomes. Thus, hypertension can be both a cause and a consequence of CKD [10, 11]. Factors that directly cause kidney damage, increase susceptibility to CKD, worsen kidney damage, and lead to a more rapid decline in kidney function include old age, family history, diabetes mellitus (DM), high blood pressure, poor glycemic control, and smoking [8]. The synergy between CKD and cardiovascular diseases threats amplified the morbidity and consequently poor outcomes with significant health and socioeconomic impact [12].

Effective screening and self-management of hypertension require patient awareness of the potential risks this condition poses for developing CKD [7]. Knowledge and lifestyle modifications are pivotal to the prevention of CKD and its progression of CKD [13, 14]. Studies show that patients' capacity to slow the progression of CKD may be limited by their lack of knowledge about the disease [15]. Although some studies on CKD have been conducted in Tanzania, those focusing on populations at risk of CKD, particularly regarding CKD knowledge and preventive practices among patients with hypertension, are scarce. One previous study in northern Tanzania's Kilimanjaro region was community-based and focused only on CKD prevalence and awareness [4]. Another study in the same region developed and validated a general survey tool for evaluating CKD knowledge, attitudes, and practices [16], but it had limited questions on CKD knowledge and preventive practices, making it unsuitable for our study. Therefore, this study aimed to assess context-specific factors associated with CKD knowledge and preventive practices among patients with hypertension in urban Dar es Salaam, specifically at Amana Regional Referral Hospital.

## 2. Methods and methods

### 2.1. Study design

This study used a quantitative approach with analytical cross-sectional design to assess factors associated with knowledge and preventive practice on CKD among hypertensive patients. This approach was considered appropriate as we wanted to ascertain the factors associated with the knowledge and preventive practice on CKD [17, 18].

### 2.2. Study setting

This study was conducted at Amana Regional Referral Hospital (RRH) in Dar es Salaam, Tanzania. Dar es Salaam is the largest and fastest-growing commercial city in Tanzania, with a population of around 5.4 million [19, 20]. Amana RRH is one of the 28 Regional Referral Hospitals and also serves as a teaching hospital within the Dar es Salaam City Council. The hospital has a total bed capacity of 362 and provides both inpatient and outpatient services, including specialized clinics in obstetrics and gynecology, neonatal and child health, ophthalmology, ENT, orthopedics, and internal medicine. During data collection, outpatient services for hypertension were provided twice a week, on Tuesdays and Thursdays, from 8:00 AM to 3:00 PM. On average, the hypertension clinic served 35 patients per day, staffed with 7 personnel, including 3 doctors and 4 nurses.

### 2.3. Study population

The study population comprised patients with hypertension aged 18 years and above, attending the hypertensive clinics at Amana RRH for care and treatment. We excluded patients who were acutely ill and needed emergency medical services.

### 2.4. Sample size and sampling procedure

The sample size was estimated using the cross-sectional formula for a study when the population size is known [21], resulting in a required sample size of 184. Based on the sampling frame established from the daily patients' appointments at the clinic, a simple random sampling technique was employed to recruit respondents until the sample size was reached. During the selection process, two pieces of paper marked "YES" and "NO" were folded and placed in a box. Potential respondents were asked to pick one paper at a time without repeating.

Those who selected a paper marked "YES" were included in the study. This technique ensured an equal chance of selecting the respondents.

## 2.5. Data collection tools and procedure

**2.5.1. Data collection tools.**   Data was collected using a questionnaire adapted from a previous study conducted in Jordan [13], comprising three sections. The first section covered sociodemographic characteristics, including age, sex, education level, and employment status. The second section assessed CKD knowledge with 24 items, addressing general knowledge of CKD, kidney functions, tests used to determine kidney health, CKD risk factors, and signs and symptoms of CKD. The items were measured on a trichotomous scale ("True" or "False", or "I don't know"), asking patients to identify the correct answers, with the "I don't know" response considered incorrect. The third section evaluated CKD preventive practices with 9 items, measured on a dichotomous scale ("Yes" or "No"), recording each patient's healthy practices towards CKD prevention. The tool was translated from English into Swahili, the most widely spoken language in Tanzania, and was piloted within the same study population and setting, with data from the pilot phase excluded from the final analysis. To determine the number of participants for the pilot test, we used 10% of the calculated sample size. The purpose of the pilot was to pretest the tool by assessing the accuracy, consistency, and relevance of the questions. Based on the feedback from pretest, the tool was revised and modified. A reliability test for the tool using Cronbach's alpha produced a coefficient of 0.798, indicating good reliability.

**2.5.2. Data collection procedure.**   Data collection for this study was conducted from 5[th] June to 20[th] July, 2022. The researchers approached potential respondents with the assistance of staff nurses from the hypertensive clinic. They explained the aim of the study and requested consent. Self-administered questionnaires were then distributed to respondents to read and complete. Researchers conducted face-to-face interviews with some respondents who needed assistance to complete the questionnaire due to difficulties in filling caused by old age. Questionnaires were filled out while respondents waited to see healthcare providers or afterward, in a nearby room ensuring privacy for face-to-face interviews. Throughout the process, researchers were available for clarification and supported respondents in completing the questionnaire, checking each questionnaire upon completion. Data collection was facilitated by four research assistants who were registered nurses trained over three days to familiarize themselves with the study's purpose and data collection procedures. This included training them to ensure that both self-administered and interviewer-administered questionnaires had clear, consistent instructions for respondents to maintain uniformity in how questions were understood and answered.

**2.5.3. Measures.**   Knowledge about CKD was assessed using a median score calculated from 24 items. Respondents were categorized as having high knowledge if they scored $\geq$ the median score and low knowledge if they scored $<$ the median score. The same approach was used to assess specific CKD knowledge domains i.e. functions of the kidney, tests that can determine the health of kidneys, risk factors for CKD, and signs and symptoms CKD. Similarly, CKD preventive practices (aimed at limiting CKD development) were assessed using a median score derived from 9 items. Respondents were categorized as practicing good preventive measures if they scored $\geq$ the median score and poor preventive measures if they scored $<$ the median score. These measurement methods for knowledge and preventive practices regarding CKD were adapted from previous studies, with Cronbach's alpha of between 0.62 and 0.87 [13, 22, 23].

## 2.6. Statistical analysis

Data analysis was conducted using IBM-SPSS Statistics version 25. Categorical variables were presented as frequencies (percentages), while continuous variables were reported as median (interquartile ranges (IQR)) means ± standard deviation (SD). The Kolmogorov—Smirnov test was employed to assess normality of continuous data. Variables not following a normal distribution were described using medians and their IQR. Bivariate logistic regression analysis was conducted to identify factors associated with knowledge and preventive practices related to CKD. Subsequently, multiple logistic regression analysis was performed, including factors with p-values $\leq 0.2$ from the bivariate analysis, to further ascertain these associations. In all analyses, statistical significance was set at a p-value $< 0.05$.

## 2.7. Ethical consideration

Ethical clearance for this study was granted by the Research Ethics Committee of Muhimbili University of Health and Allied Sciences (MUHAS) with reference number DA.282/298/01.C/ 1192. Permission to conduct the study was also obtained from the Medical Officer In-charge of Amana RRH. Written informed consent was obtained from all participants before their involvement in the study. Participants were informed of their right to withdraw from the study at any time, and their participation was entirely voluntary. Confidentiality was strictly maintained by ensuring the anonymity of all documents containing participants' information.

## 3. Results

### 3.1. Respondents' sociodemographic characteristics

One hundred and eighty-four (184) respondents completed the questionnaire without any missing values. The median (IQR) age was 62 (54–70) years, with more than half of the respondents aged 61 and above. Females constituted the majority (74.5%) of the respondents. Regarding education, the vast majority (82.6%) had either primary or secondary level of education. Approximately half of the respondents (53.8%) were either employed or self-employed. Table 1 provides a summary of the sociodemographic characteristics of the respondents.

### 3.2. Knowledge of CKD and associated factors

The median (IQR) score for CKD knowledge was 13 (9–16). Respondents scoring $\geq 13$ were classified as having high knowledge, while those scoring $< 13$ were classified as having low knowledge. Based on this criterion, 104 (56.5%) respondents demonstrated high CKD knowledge, while 80 (43.5%) had low CKD knowledge. In terms of specific CKD knowledge domains, 112 (60.9%) of patients had high knowledge of kidney functions, 107 (58.2%) on

**Table 1. Sociodemographic characteristics of respondents (N = 184).**

| Variable | Category | Frequency (n) | Percentage (%) |
|---|---|---|---|
| Age groups (years) | 18–60 | 83 | 45.1 |
| | $\geq 61$ | 101 | 54.9 |
| Gender | Female | 137 | 74.5 |
| | Male | 47 | 25.5 |
| Level of education | Primary/Secondary | 152 | 82.6 |
| | College/University | 32 | 17.4 |
| Employment status | Unemployed | 85 | 46.2 |
| | Employed | 99 | 53.8 |

**Table 2. Frequency distribution of respondent responses to CKD knowledge questions.**

| Items | Correct n (%) | Incorrect n (%) |
|---|---|---|
| **General knowledge of CKD:** | | |
| A person can lead a normal life with one healthy kidney | 84 (45.7) | 100 (54.3) |
| Herbal supplements can be effective in treating CKD | 113 (61.4) | 71 (38.6) |
| Some medications can help to slow-down the worsening of CKD | 137 (74.5) | 47 (25.5) |
| **Functions of the kidney:** | | |
| The kidneys make urine | 154 (83.7) | 30 (16.3) |
| The kidneys clean blood | 95 (51.6) | 89 (48.4) |
| The kidneys help to keep blood sugar level normal | 96 (52.2) | 88 (47.8) |
| The kidneys help to maintain blood pressure | 89 (48.4) | 95 (51.6) |
| The kidneys help to breakdown protein in the body | 33 (17.9) | 151 (82.1) |
| The kidneys help to keep the bones healthy | 87 (47.3) | 97 (52.7) |
| **Tests that can determine the health of your kidneys:** | | |
| Blood test | 107 (58.2) | 77 (41.8) |
| Urine test | 89 (48.4) | 95 (51.6) |
| Fecal (poo) test | 64 (34.8) | 120 (65.2) |
| Blood pressure monitoring | 45 (24.5) | 139 (75.5) |
| **Risk factors for CKD:** | | |
| Diabetes | 128 (69.6) | 56 (30.4) |
| Being female | 87 (47.3) | 97 (52.7) |
| High blood pressure | 121 (65.8) | 63 (34.2) |
| Heart problems such as heart failure or heart attack | 125 (67.9) | 59 (32.1) |
| Excess stress | 88 (47.8) | 96 (52.2) |
| Obesity | 110 (59.8) | 74 (40.2) |
| **Signs and symptoms CKD:** | | |
| Water retention | 139 (75.5) | 45 (24.5) |
| Fever | 72 (39.1) | 112 (60.9) |
| Nausea/vomiting | 71 (38.6) | 113 (61.4) |
| Loss of appetite | 110 (59.8) | 74 (40.2) |
| Fatigue | 123 (66.8) | 61 (33.2) |

tests to assess kidney health, 106 (57.6%) on CKD risk factors, and 109 (59.2%) on the signs and symptoms of CKD.

Table 2 presents the frequency distribution of respondents' responses to CKD knowledge questions. The item with the highest knowledge score was understanding the function of kidneys in urine production (n = 154; 83.7%), followed by awareness that certain medications can slow the progression of CKD (n = 137; 74.5%). Conversely, the item with the lowest CKD knowledge score was the misconception that kidneys help break down proteins in the body, with more than four-fifths of respondents (n = 151; 82.1%) holding this misconception. Additionally, about three-quarters of respondents (n = 139; 75.5%) were unaware that blood pressure tests can assess kidney health.

In bivariate and multiple logistic regression analyses (Table 3), none of the sociodemographic factors showed a significant association with CKD knowledge.

## 3.3. Preventive practice for CKD and associated factors

The median (IQR) score for CKD preventive practices was 9 (8–9). Respondents scoring ≥ 9 were classified as having good preventive practices, while those scoring < 9.00 were classified

**Table 3. Factors associated with knowledge on CKD (N = 184).**

| Variables | Low knowledge (median <13) n (%) | High Knowledge (median ≥13) n (%) | Knowledge of CKD | | | |
|---|---|---|---|---|---|---|
| | | | Unadjusted COR (95% CI) | $p$-value | Adjusted AOR (95% CI) | $p$-value |
| **Age** | | | | | | |
| 18–60 | 35 (42.2) | 48 (57.8) | Ref | | – | |
| ≥ 61 | 45 (44.6) | 56 (55.4) | 0.91 (0.51, 1.63) | 0.745 | – | – |
| **Gender** | | | | | | |
| Female | 55 (40.1) | 82 (59.9) | Ref | | Ref | |
| Male | 25 (53.2) | 22 (46.8) | 0.59 (0.30, 1.15) | 0.121 | 0.62 (0.32, 1.23) | 0.171 |
| **Level of Education** | | | | | | |
| Primary/ Secondary | 66 (43.4) | 86 (56.6) | Ref | | – | |
| College/University | 14 (43.8) | 18 (56.2) | 0.99 (0.46, 2.13) | 0.973 | – | – |
| **Employment status** | | | | | | |
| Unemployed | 42 (49.4) | 43 (50.6) | Ref | | Ref | |
| Employed | 38 (38.4) | 61 (61.6) | 1.57 (0.87, 2.82) | 0.133 | 1.49 (0.82, 2.70) | 0.188 |

Abbreviations: COR = crude odds ratio, AOR = adjusted odds ratio, CI = confidence interval, Hosmer-Lemeshow test = 0.559.

as having poor preventive practices. Among respondents, 111 (60.3%) demonstrated good preventive practices for CKD.

Table 4 presents the frequency distribution of respondents' responses to CKD preventive practice questions. Five items were highly scored by respondents (98.9%), including regular hypertension screening, incorporating vegetables into meals, adherence to hypertension treatment, abstaining from alcohol, and non-smoking. An item with the lowest score was regular renal checkups as advised healthcare providers, with 125 respondents (68.4%) reporting adherence to this practice.

Table 5 shows the bivariate and multiple logistic regression analyses. In the bivariate analysis, high CKD knowledge (COR: 1.96; 95% CI: 1.08, 3.57; $p$ = 0.028) was significantly associated with good CKD preventive practices. In the multiple logistic regression analysis, only high CKD knowledge (AOR: 1.98; 95% CI: 1.08, 3.62; $p$ = 0.027) remained significantly associated with good CKD preventive practices.

**Table 4. Frequency distribution of respondent responses to CKD preventive practice.**

| Items | Yes n (%) | No n (%) |
|---|---|---|
| Do you have regular physical exercise at least three times a week? | 181 (98.4) | 3 (1.6) |
| Do you have regular renal check-up as advised your healthcare providers? | 125 (67.9) | 59 (32.1) |
| Do you have regular hypertension screen in every week? | 182 (98.9) | 2 (1.1) |
| Do you regularly include vegetables in your meals | 182 (98.9) | 2 (1.1) |
| Do you take traditional medicine without physician recommendation? | 9 (4.9) | 175 (95.1) |
| Do you regularly follow food restrictions, such as low salt diet? | 174 (94.6) | 10 (5.4) |
| Do you follow hypertension medication regimen/ treatment adherence? | 182 (98.9) | 2 (1.1) |
| Do you drink more than two bottles of alcohol per day (for women >2 and men >3 per day)? | 2 (1.1) | 182 (98.9) |
| Do you smoke cigarette? | 2 (1.1) | 182 (98.9) |

**Table 5. Factors associated with CKD preventive practice (N = 184).**

| Variables | Low knowledge (median <13) n (%) | High Knowledge (median ≥13) n (%) | Knowledge of CKD | | | |
|---|---|---|---|---|---|---|
| | | | Unadjusted COR (95% CI) | p-value | Adjusted AOR (95% CI) | p-value |
| **Age** | | | | | | |
| 18–60 | 31 (37.3) | 52 (62.7) | Ref | | – | |
| ≥ 61 | 42 (41.6) | 59 (58.4) | 0.83 (0.46, 1.52) | 0.559 | – | – |
| **Gender** | | | | | | |
| Female | 54 (39.4) | 83 (60.6) | Ref | | – | |
| Male | 19 (40.4) | 28 (59.6) | 0.96 (0.49, 1.89) | 0.903 | – | – |
| **Level of Education** | | | | | | |
| Primary/ Secondary | 64 (42.1) | 88 (57.9) | Ref | | Ref | |
| College/University | 9 (28.1) | 23 (71.9) | 1.86 (0.81, 4.28) | 0.146 | 1.89 (0.81, 4.41) | 0.139 |
| **Employment status** | | | | | | |
| Unemployed | 33 (38.8) | 52 (61.2) | Ref | | – | |
| Employed | 40 (40.4) | 59 (59.6) | 0.94 (0.52, 1.69) | 0.827 | – | – |
| **CKD knowledge** | | | | | | |
| Poor | 39 (48.8) | 41 (51.2) | Ref | | Ref | |
| Good | 34 (32.7) | 70 (67.3) | 1.96 (1.08, 3.57) | 0.028 | 1.98 (1.08, 3.62) | 0.027 |

Abbreviations: COR = crude odds ratio, AOR = adjusted odds ratio, CI = confidence interval, Hosmer-Lemeshow test = 0.363.

## 4. Discussion

Research evidence consistently shows that knowledge and lifestyle modifications are essential in preventing and managing the progression of CKD [13, 14]. This study assessed factors associated with CKD knowledge and preventive practices among patients with hypertension in Dar es Salaam. Our results revealed that more than half of the patients demonstrated high CKD knowledge and engaged in good CKD preventive practices. Notably, a positive association was observed between CKD knowledge and preventive practices. These results emphasize the importance of educational interventions aimed at improving CKD knowledge among hypertensive individuals, which could lead to CKD preventive practices.

The median age of patients in our study was in the older, reflecting the well-established link between aging and increased prevalence of hypertension. Similar findings have been reported in studies from Ethiopia and Palestine, where hypertension is more prevalent in older age groups [13, 23]. This underscores the importance of regular hypertension check-ups in older populations, enabling early detection and timely interventions to prevent complications such as CKD.

While a significant proportion of patients exhibited high CKD knowledge, over two-fifths had low knowledge. Specifically, between 57.6% and 60.9% of patients demonstrated a strong understanding of kidney functions, CKD risk factors, CKD signs and symptoms, and relevant diagnostic tests. However, knowledge gaps were evident in specific areas: over 80% mistakenly believed that kidneys are involved in breaking down proteins, and nearly 75% were unaware that blood pressure levels are a key indicator of kidney health. These misconceptions and gaps in knowledge may hinder CKD prevention efforts, as controlling blood pressure is essential for kidney health [13]. Literature shows that understanding target blood pressure levels is linked to improved control, and consequently, better CKD outcomes [24]. This highlights the need for focused educational efforts to address such gaps, particularly in high-risk populations

and LMICs, where the burden of CKD is disproportionally high and strains healthcare systems [25–28].

The level of CKD knowledge observed in our study is comparable to that in Rwanda, where more than half of university students had high CKD knowledge despite differences in study populations [29]. However, CKD knowledge in our cohort was higher than in studies from Ethiopia and Congo Brazzaville, where fewer than half of hypertensive patients had high CKD knowledge [23, 30]. Conversely, other studies from Ethiopia and Kenya reported much higher knowledge levels, with more than two-thirds of hypertensive patients exhibiting high CKD knowledge [31, 32]. These variations in CKD knowledge across LMICs can be attributed to differences in healthcare access, socioeconomic disparities, cultural beliefs, healthcare system capacities, and policy implementations [33, 34]. Such disparities reinforce the need for context-specific educational programs to address local barriers to knowledge dissemination.

Interestingly, none of the sociodemographic characteristics, including age, gender, level of education, and employment status, were significantly associated with CKD knowledge in our study. This suggests that sociodemographic factors did not play a major role in knowledge acquisition among these patients. While our findings align with some studies, others research presents conflicting results. For instance, in Ethiopia age was not linked to CKD knowledge, but higher education was [23]. Similarly, a study from Palestine showed that while age, gender, and employment status did not predict knowledge, younger age and higher education were associated with better CKD knowledge [13]. These disparities highlight the complexity of CKD knowledge distribution across LMICs, necessitating targeted educational strategies based on local contexts [31, 33, 34].

Regarding CKD preventive practices, approximately three-fifths of patients engaged in good preventive lifestyle modifications, including regular hypertension screening, dietary adjustments (e.g., increased vegetable intake), adherence to hypertension treatment, and avoiding alcohol and smoking. However, two-fifths did not follow these preventive measures, reflecting significant gaps in adherence to lifestyle changes essential for CKD prevention. Socioeconomic factors previously reported in Tanzania, such as financial constraints, healthcare costs, and logistical barriers (e.g., transportation issues), likely contributed to this low adherence [35]. Similar trends of poor preventive practices have been observed in Ethiopia, where fewer than half of hypertensive patients adhered to CKD preventive measures [32]. Conversely, in Kenya, over 80% of hypertensive patients engaged in good preventive practices, highlighting regional differences [31]. These variations may stem from factors such as healthcare access, cultural beliefs, and socioeconomic conditions, supporting the need for tailored interventions that address local challenges.

Remarkably, none of the sociodemographic factors in our study were associated with CKD preventive practices. This contrasts with findings from Kenya, where female patients with hypertension and diabetes were more likely to engage in preventive practices [31]. In Palestine, male patients exhibited higher preventive practices, while in Ethiopia, higher education predicted better preventive behavior [13, 32]. Such findings suggest that factors beyond sociodemographics, such as health system functionality, and socioeconomic conditions, might have a more significant impact on preventive practices [31].

Crucially, our study found that high CKD knowledge was associated with a twofold increase in the likelihood of engaging in CKD prevention. This finding aligns with previous research in Kenya and Palestine, which also reported that higher CKD knowledge was linked to better preventive practices among hypertensive patients [13, 31]. These results underscore the importance of education in improving lifestyle choices and fostering CKD prevention [36, 37]. Tailored educational programs, delivered in both healthcare facilities and community settings, are essential for improving knowledge and encouraging preventive behaviors,

especially in LMICs like Tanzania, where advanced CKD care is largely inaccessible and unaffordable [27].

Several limitations should be considered in the current study. First, social desirability bias may have influenced patients' responses, particularly regarding their knowledge and preventive behaviors, potentially leading to an overestimation of both. Second, the study's urban setting limits the generalizability of our findings to rural populations, where access to healthcare and socioeconomic conditions differ. Third, the cross-sectional design precludes establishing causality between CKD knowledge and preventive practices, although there is overarching relationship between the two as they influence each other, as established in literature. Lastly, we did not assess potential confounders, such as healthcare access, socioeconomic status and comorbidities, which could have influenced our findings. Despite these limitations, the study provides valuable insights into CKD knowledge and preventive practices among hypertensive patients in an urban Tanzanian setting, which can inform future interventions.

## 5. Conclusions

Although hypertensive patients in this study exhibited high CKD knowledge and engaged in preventive practices, a significant proportion did not. Our findings underscore the positive relationship between CKD knowledge and preventive behaviors, highlighting the crucial role of educational interventions. Targeted, multifaceted programs, including both individual and group sessions, are crucial for improving self-management and health outcomes [26]. In resource-limited settings like Tanzania, nurse-led health educational programs delivered both in healthcare facilities and communities can play a key role in correcting misconceptions and promoting regular health check-ups, hypertension monitoring, and renal screenings [27, 28]. Future research should expand to rural areas in Tanzania and address potential confounders to offer a more comprehensive understanding of factors influencing CKD knowledge and preventive practices among hypertensive patients.

## Supporting information

**S1 Dataset.**
(SAV)

## Acknowledgments

We would like to thank the Amana Regional Referral Hospital for permitting us to conduct this study. We specifically appreciate the time and valuable information provided by patients with hypertension that made this study a success.

## Author Contributions

**Conceptualization:** Shabani S. Ngulupi.

**Formal analysis:** Joel Seme Ambikile, Agnes Fredrick Massae.

**Funding acquisition:** Shabani S. Ngulupi.

**Investigation:** Shabani S. Ngulupi.

**Methodology:** Joel Seme Ambikile, Agnes Fredrick Massae.

**Supervision:** Joel Seme Ambikile, Agnes Fredrick Massae.

**Validation:** Agnes Fredrick Massae.

**Writing – original draft:** Joel Seme Ambikile.

**Writing – review & editing:** Shabani S. Ngulupi, Agnes Fredrick Massae.

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
