## [Decision Letter · Decision Letter 0]

9 Sep 2024

PONE-D-24-30797Factors Associated with Chronic Kidney Disease Knowledge and Preventive Practices: An Analytical Cross-sectional Study Among Patients with Hypertension at Amana Regional Referral Hospital in Dar es Salaam, TanzaniaPLOS ONE

Dear Dr. Ambikile,

Thank you for submitting your manuscript to PLOS ONE. After careful consideration, we feel that it has merit but does not fully meet PLOS ONE’s publication criteria as it currently stands. Therefore, we invite you to submit a revised version of the manuscript that addresses the points raised during the review process.

We look forward to receiving your revised manuscript.

Kind regards,

Tauqeer Hussain Mallhi, Ph.D

Academic Editor

PLOS ONE

“This study was funded by the Ministry of Health in Tanzania”

Additional Editor Comments:

Thank you for your submission to PLOS ONE. The manuscript has been reviewed by two relevant experts in the field. In addition to their feedback, I have a few additional comments that the authors are required to address:

The authors could add information on the reliability of the Kiswahili version of the questionnaire if data is available from the pilot study. Reporting Cronbach’s alpha or any validation data would improve confidence in the tools used.

In the discussion, the authors could acknowledge potential confounding factors, such as access to healthcare and socioeconomic status, that may have influenced CKD knowledge and preventive practices. This would demonstrate an understanding of factors that could be addressed in future studies.

Reanalyzing the existing dataset could provide a more detailed interpretation of knowledge and preventive practices. Breaking down knowledge into specific domains (e.g., kidney function vs. CKD symptoms) could identify where the gaps in understanding are most prevalent.

The discussion section could be expanded to speculate on reasons for low adherence to preventive practices, such as regular renal checkups. The authors could reference existing literature on healthcare access or cultural factors in Tanzania that may influence health-seeking behavior.

The recommendations for educational programs could be strengthened by suggesting more targeted approaches, such as focusing on correcting specific misconceptions or emphasizing the importance of regular renal checkups, supported by references to similar interventions.

The study’s limitations, such as the cross-sectional design and inability to establish causality, should be explicitly discussed to enhance the integrity of the conclusions.

If supplementary tables or additional analyses were conducted but not included, these could be added to provide further insights without the need for new data collection.

Reviewers' comments:

Reviewer's Responses to Questions

**Comments to the Author**

1. Is the manuscript technically sound, and do the data support the conclusions?

Reviewer #1: Yes

Reviewer #2: Partly

2. Has the statistical analysis been performed appropriately and rigorously? 

Reviewer #1: Yes

Reviewer #2: Yes

3. Have the authors made all data underlying the findings in their manuscript fully available?

Reviewer #1: Yes

Reviewer #2: Yes

4. Is the manuscript presented in an intelligible fashion and written in standard English?

Reviewer #1: Yes

Reviewer #2: Yes

5. Review Comments to the Author

Reviewer #1: Ambikile et al present an interesting study that used patient questionnaires to assess factors associated with CKD knowledge and preventive practices among patients with hypertension in Dar es Salaam. The results revealed that >50% of study patients had high CKD knowledge and good CKD preventive practices. Importantly, higher CKD knowledge was associated with better preventive practices. The paper is well written with minor typos and grammatical errors. The paper is generalizable and the authors do a good job comparing their results to that of similar studies from other African and middle eastern countries.

1. Could you elucidate more on the comorbidities that the study patients have? Ideally, this should be on table 1. The level of comorbidities may also influence patient knowledge and their motivation for CKD preventive measures.

2. The study could be strengthened by showing the association between CKD knowledge and also blood pressure control or medication adherence. Do you have data on the blood pressure management of the patients in the study?

Reviewer #2: Dear Authors,

I have reviewed your submitted manuscript titled "Factors Associated with Chronic Kidney Disease Knowledge and Preventive Practices: An Analytical Cross-sectional Study Among Patients with Hypertension at Amana Regional Referral Hospital in Dar es Salaam, Tanzania" and have the following comments:

Introduction

Lines 72-73 - kindly reference the KDIGO sentence.

Methods

Line 109 - be exact with the number of beds; it is a finite number and will be known.

Line 141- Please clarify which cohort the excluded patients came from? Were they from the sample population i.e. the 184 patients or from the population of patients that had not met study eligibility ab initio (those who had chosen NO at simple randomization)? If from the sample population, could the patients who selected NO at random sampling not have been used as the pilot group? The exclusion of the data of the study sample population will reduce the effective sample size and the subsequent statistical analyses based on the downsized numbers.

Results

Lines 291, 222 - the words "endorse", "endorsement" are conveying ambigous meanings with respect to the responses to the direct questions as itemized in Table 4. What do the authors mean by endorsement? Acknowledgment of the knowledge of the prevenatitive measure or an ascertainment of the actual practice of preventative measures? Kindly find a word to express what is meant in these lines.

Lines 219, 222 and Table 4 - the authors report that 68.4% of respondents engaged in 3 monthly renal check ups. Why were they doing this? Is this routine practice/standard of care in your hypertension clinic?

Discussion

The discussion needs to be more robust and not just a comparison of your findings with the findings of others. What are the implications of the similarities/differences of your results relative to others' results for your patient population, standard of care practices, possible policy establishment or change etc. For example lines 250-251 can be discussed further as "about three-quarters were unaware that blood pressure levels can indicate kidney health." From a public health perspective, this is a very significant/concerning finding that should be discussed in an at-risk population in a world geographical region where preventative measures for CKD are to be emphasised and promoted.

Limitations

Line 318 -319 - "the cross-sectional nature of the study prevents us from establishing a causal relationship between sociodemographic characteristics, CKD knowledge, and preventive practices." I do not believe that that there is a scientific need for causality in your study. The overaching relationship between knowledge and practice will be an association; knowledge or lack therefore influences practice but does not cause practice in a scientific sense.

General comments: a few grammatical errors through the manuscript; kindly have an independent person read through. The authors not likely to pick them due to familiarity with the text.

Thank you.

6. PLOS authors have the option to publish the peer review history of their article (what does this mean?). If published, this will include your full peer review and any attached files.

Reviewer #1: No

Reviewer #2: No

---

## [Author Response · Author response to Decision Letter 0]

26 Sep 2024

Specific reviewer and editor comments have been addreesed in the response to reviewers comments, a file attached in this submission

---

## [Decision Letter · Decision Letter 1]

25 Dec 2024

Factors Associated with Chronic Kidney Disease Knowledge and Preventive Practices: An Analytical Cross-sectional Study among Patients with Hypertension at Amana Regional Referral Hospital in Dar es Salaam, Tanzania

PONE-D-24-30797R1

Dear Dr. Ambikile,

We’re pleased to inform you that your manuscript has been judged scientifically suitable for publication and will be formally accepted for publication once it meets all outstanding technical requirements.

Kind regards,

Ken Iseri

Academic Editor

PLOS ONE

Additional Editor Comments (optional):

The authors have addressed the major concerns of the reviewers.

Reviewers' comments:

Reviewer's Responses to Questions

**Comments to the Author**

1. If the authors have adequately addressed your comments raised in a previous round of review and you feel that this manuscript is now acceptable for publication, you may indicate that here to bypass the “Comments to the Author” section, enter your conflict of interest statement in the “Confidential to Editor” section, and submit your "Accept" recommendation.

Reviewer #3: All comments have been addressed

Reviewer #4: All comments have been addressed

2. Is the manuscript technically sound, and do the data support the conclusions?

Reviewer #3: Partly

Reviewer #4: Yes

3. Has the statistical analysis been performed appropriately and rigorously? 

Reviewer #3: Yes

Reviewer #4: I Don't Know

4. Have the authors made all data underlying the findings in their manuscript fully available?

Reviewer #3: Yes

Reviewer #4: Yes

5. Is the manuscript presented in an intelligible fashion and written in standard English?

Reviewer #3: Yes

Reviewer #4: Yes

6. Review Comments to the Author

Reviewer #3: As the study population seems to be mostly educated people, there may be a bias in the conclusions which may not be applicable to the general population. This may be added as a possible limitation of the study

Reviewer #4: Thank you for addressing the comments from the reviewers.

It is very interesting to learn about the level of CKD awareness in your study population.

7. PLOS authors have the option to publish the peer review history of their article (what does this mean?). If published, this will include your full peer review and any attached files.

Reviewer #3: **Yes: **Jacob George

Reviewer #4: No

---

## [Editor Report · Acceptance letter]

29 Dec 2024

PONE-D-24-30797R1 

PLOS ONE

Dear Dr. Ambikile, 

I'm pleased to inform you that your manuscript has been deemed suitable for publication in PLOS ONE. Congratulations! Your manuscript is now being handed over to our production team.

Kind regards, 

on behalf of

Dr. Ken Iseri 

Academic Editor

PLOS ONE